# Porphyrin Based 2D-MOF Structures as Dual-Kinetic Sorafenib Nanocarriers for Hepatoma Treatment

**DOI:** 10.3390/ijms222011161

**Published:** 2021-10-16

**Authors:** Adam Bieniek, Marek Wiśniewski, Joanna Czarnecka, Jędrzej Wierzbicki, Marcin Ziętek, Maciej Nowacki, Dariusz Grzanka, Tomasz Kloskowski, Katarzyna Roszek

**Affiliations:** 1Physicochemistry of Carbon Materials Research Group, Faculty of Chemistry, Nicolaus Copernicus University in Toruń, 87-100 Toruń, Poland; abieniek@umk.pl; 2Department of Biochemistry, Faculty of Biological and Veterinary Sciences, Nicolaus Copernicus University in Toruń, 87-100 Toruń, Poland; j_czar@umk.pl; 3Student’s Scientific Society, Ludwik Rydygier Collegium Medicum in Bydgoszcz, Nicolaus Copernicus University in Torun, Jagiellońska Street 13/15, 85-067 Bydgoszcz, Poland; jedrzejwierzbicki@gmail.com; 4Department of Oncology, Wroclaw Medical University, 53-413 Wroclaw, Poland; zietekm@op.pl (M.Z.); maciej.s.nowacki@gmail.com (M.N.); 5Lower Silesian Comprehensive Cancer Center, Department of Surgical Oncology, Hirszfelda 12, 53-413 Wroclaw, Poland; 6Department of Dermatology, Sexually Transmitted Diseases and Immunodermatology, Ludwik Rydygier Medical College in Bydgoszcz, Nicolaus Copernicus University in Torun, Sklodowskiej-Curie 9 Street, 85-094 Bydgoszcz, Poland; 7Department of Pathology, Faculty of Medicine, Collegium Medicum in Bydgoszcz, Nicolaus Copernicus University in Torun, Sklodowskiej-Curie 9 Street, 85-094 Bydgoszcz, Poland; d_grzanka@cm.umk.pl; 8Department of Regenerative Medicine, Cell and Tissue Bank, Nicolaus Copernicus University in Torun, Ludwik Rydygier Medical College in Bydgoszcz, 85-094 Bydgoszcz, Poland; tomasz.kloskowski@cm.umk.pl

**Keywords:** 2D metal–organic framework, PPF structure, multikinase inhibitors, sorafenib, anticancer drug delivery

## Abstract

The existing clinical protocols of hepatoma treatment require improvement of drug efficacy that can be achieved by harnessing nanomedicine. Porphyrin-based, paddle-wheel framework (PPF) structures were obtained and tested as dual-kinetic Sorafenib (SOR) nanocarriers against hepatoma. We experimentally proved that sloughing of PPF structures combined with gradual dissolving are effective mechanisms for releasing the drug from the nanocarrier. By controlling the PPF degradation and size of adsorbed SOR deposits, we were able to augment SOR anticancer effects, both in vitro and in vivo, due to the dual kinetic behavior of SOR@PPF. Obtained drug delivery systems with slow and fast release of SOR influenced effectively, although in a different way, the cancer cells proliferation (reflected with EC50 and ERK 1/2 phosphorylation level). The in vivo studies proved that fast-released SOR@PPF reduces the tumor size considerably, while the slow-released SOR@PPF much better prevents from lymph nodes involvement and distant metastases.

## 1. Introduction

Hepatocellular carcinoma (HCC) is one of the deadliest cancers due to its complexity, reoccurrence after surgical resection, metastasis and heterogeneity. New cases are diagnosed annually in over 500,000 patients worldwide, and it is the second leading cause of cancer death in the world [1]. Incidence and mortality of HCC are increasing in Western European countries and are expected to rise as a consequence of the obesity epidemic. Multiple factors trigger the initiation and progression of HCC, including chronic alcohol consumption, viral hepatitis B and C infection, metabolic disorders and age. Hepatic cancer therapies are currently limited to surgery, radiation, and chemotherapy, but all these methods risk damage to normal tissues or incomplete eradication of the cancer. Therefore, there is a constant search for more and more effective therapies that require even more complex, advanced drugs, consisting of active compounds and tailored drug delivery systems (DDS). Although various multikinase inhibitors have been tested as systemic therapies of HCC, only sorafenib and lenvatinib (orally active multi-targeted tyrosine kinase inhibitors) deserve special mention as FDA-approved drugs for the treatment of advanced HCC [2,3]. Sorafenib (SOR) was originally developed as an inhibitor of serine/threonine Raf kinases, which are known to be important in tumor cell signaling. Accordingly, this drug has been shown to have the activity of multikinase inhibitor targeting several receptor tyrosine kinases, including VEGFR-2 and PDGFR-β, and their associated signaling cascades of the ERK pathway, involved in tumor growth and angiogenesis [4]. Contemporary anticancer therapy with the use of orally administered SOR suffers from the drawbacks of dose-limiting toxicities, development of multi-drug resistance (MDR) and unfavorable side-effects as fatigue, diarrhea, hypertension, skin toxicity, weight loss, and hypophosphatemia. Therefore, SOR urgently requires improvement in its effectiveness [3] offered inter alia by nanomedicine. One of the ways to increase drug efficacy is the use of DDS/nanocontainers with an internalized anticancer drug, which will create new possibilities of local drug delivery and controlled release, while reducing side effects.

Drug delivery systems with SOR used so far were nanopolymer systems, i.e., polylactic acid (PLA) and poly-(lactic-co-glycolic) acid (PLGA), polyethylene glycol (PEG), lipo-somes, solid lipid nanoparticles, nanostructured lipid carriers, as well as silica nanoparticles and carbonaceous nanostructures (CNT, graphene oxide). In addition, attempts were made to extend the capabilities of the obtained SOR@nanocarrier with targeted, pH-responsive, and magnet-responsive properties, as well as in combination therapies with other chemotherapeutics (e.g., Doxorubicin), siRNA, photodynamic- and phototermal therapy (PDT/PTT), or immunotherapeutics [5]. Despite their many advantages, the properties of the currently studied nanoparticles can still cause problems related to the increased friction and adhesion in blood vessels. In addition, they may be responsible for the increased production of reactive oxygen species in vivo, thus leading to cytotoxicity and unpredictable interactions [6]. The search for an optimal drug carrier is still one of the challenges in nanomedicine.

One of the recently exploited DDS are metal–organic frameworks (MOFs), which are also increasingly used in cancer therapy. MOFs are one of the most promising groups of candidates for DDS, bridging the gap between organic and inorganic materials [7,8,9], where the interactions between linkers and metals promote the gradual degradation of MOF and release of an active agent [10]. An exciting subgroup of MOF, from the point of view of anticancer therapy, are materials made of porphyrins. Porphyrins represent the oldest and most widely studied chemical structures, both in nature and in biomedical applications [11,12]. The remarkable candidate for the formation of porphyrin MOFs is TCPP (tetra-(4-carboxyphenyl) porphyrin), which has been used in many MOFs with different metallic nodes for biomedical purposes [13]. The long-term toxicity study of porphyrin-based MOF, performed by Wang et al. [14], confirmed that studied nanoparticles had extremely low systemic toxicity, and the kidneys are responsible for removing them.

To date, several porphyrin MOFs were engaged as DDS, most often using TCPP as a ligand. In 2016, Lin et al. [15] reported PCN-221 (Zr^4+^ and TCPP) as a promising oral drug carrier for methotrexate (MTX). PCN-221 exhibited low cytotoxicity toward PC12 cells, high drug loading, sustained and controlled release behavior under physiological conditions, and pH specific release without “burst effect”. In 2017, Liu et al. [16] designed a nanoscale spindle-like zirconium-porphyrin MOF (NPMOF) for FL imaging-guided synergistic chemotherapy and PDT of tumors in vivo. NPMOFs were suitable for loading the antitumor drug doxorubicin (DOX) and accumulated mainly in the circulatory system, lymph nodes, and tumor site, which contributed to the tumor therapy and inhibited cancer metastasis. The results of FL-guided synergistic chemotherapy and PDT in HepG2 tumor-bearing mice showed that the DOX-loaded NPMOFs had a satisfactory therapeutic effect and high biocompatibility to the major organs. In 2018, Zhao at al. [17] constructed a drug delivery system ZnO-gated porMOF-AS1411 (Zr^4+^ and TCPP) to efficiently deliver Doxorubicin. The above-mentioned studies indicate that MOFs composed of a zinc derivative and zinc as a node has been mainly synthesized and verified in various anticancer therapies.

Since no literature data report dual kinetics of anticancer drug release after its deposition in porphyrin-based MOFs, we decided to examine this phenomenon in the presented work. To the best of our knowledge, this is the first study determining the mechanism of deposition and release of SOR on porphyrin-based 2D MOFs—called PPF (Porphyrin Paddlewheel Framework). Adsorption and gradual, controlled desorption are of paramount importance regarding drug delivery. We have experimentally confirmed the time-dependent PPF sloughing and elucidated the mechanism of sorafenib desorption: slow- and fast-release. The obtained DDS are efficient in vitro and in vivo as anticancer drug carriers with theranostic perspectives, leaving a niche for further research.

## 2. Results and Discussion

The full characteristics of obtained porphyrin based 2D MOFs including elemental and thermogravimetric analysis, SEM, PXRD, SEM, and TEM were presented recently [18,19]. The results of these analyses are consistent with the results presented in the literature [20], and correspond to a 2D chemical structure of PPF. In the present work, we have focused on the PPF’s ability to meet the requirements for an anticancer drug delivery system.

Biomedical application of various new materials mostly requires the stability tests of studied samples. The results of our analyses are shown in Figure 1. Degradation was tested in various solutions under different conditions. The fastest PPF degradation was observed in buffer solutions containing phosphate salts—total degradation was observed in several minutes (Figure 1). Decrease in the concentration of PBS buffer to 0.0005 M caused the degradation process slowing down, 80% of PPF degraded after 6 h of immersion, being close to degradation level in pure deionized water where 75% of material degraded after 6 h immersion. A further decrease in degradation rate was observed for acetate buffer and 30% EtOH in water solutions, where the degradation reaches 56% and 3% after 6 h, respectively. It can be concluded that obtained structure decomposes gradually in time depending on the solution composition. That makes PPF beneficial for drug delivery system releasing its cargo in a controlled way.

Another important factor influencing the degradation has been temperature. From the results presented in Figure 2, one can conclude that the lower is the temperature, the slower is the degradation—down to 2.5% (after 7 h immersion) at 4 °C. Based on these results, the activation energy of degradation was calculated as 16.68 kJ/mol via assumption of 2nd order reaction kinetics.

The stability of MOF structures in water is an important issue mainly due to their multiple biomedical applications, e.g., in drug delivery systems [21,22]. Gelfand and Shimizu [23], as the first researchers, proposed six stages of a MOF exposure for the study on its stability in water. In their approach, each sample was exposed to different conditions starting from the most gentle ones, i.e., near-ambient and dry atmosphere (20 °C and 20% of RH), and finishing with the harshest conditions, i.e., in boiling water. Based on these facts, we have modified the conditions to test the PPF stability in water and applied them as follows: (i) ST-1 at 80 °C and 90% RH, (ii) ST-2 at 4 °C and immersion in H_2_O, (iii) ST-3 at 20 °C and immersion in H_2_O, and (iv) ST-4 at 100 °C and immersion in H_2_O.

The results of the analysis of the solids remaining after degradation are summarized in Figure 3. According to Figure 1 and Figure 2, the PPF degradation in water occurs undoubtedly. However, all data presented in Figure 3 confirm that only external layers are sloughed one by one during the degradation process, leaving the core of tested PPF unchanged (Figure 3A). The internal TCPP core structure is stable until the samples are not immersed in boiling water (ST-4). The XRD patterns, low temperature N_2_ adsorption isotherms (with pore size distribution), and FTIR analysis of samples treated under ST-1 to ST-3 conditions prove the prevented structure and chemistry.

The PPF structure shows four typical peaks in the XRD pattern (Figure 3B, red curve), which can be ascribed to the tetragonal structure of Zn-TCPP nodes. Since the PPF nanosheets tend to lie one on the other with (001) preferred orientation [18,20,24,25,26,27] (see also Figure 5 showing the book-pages-like overall structure), the signal at ~18° corresponding to the (004) dominates in the pattern. Moreover, this effect causes that most of the crystal diffraction peaks disappear gradually when degradation conditions become tougher. After immersion in water at 20 °C (ST-3), finally, these peaks cannot be detected, and only a broad signal of (004) plane is observed (Figure 3B, blue curve).

Such step-by-step sloughing of PPF nanosheets (schematically shown in Figure 3A) has been described and analyzed for the first time; however, it was achieved unconsciously (and not discussed) by Zhao et al. [20] while the authors were preparing 2D samples for AFM microscopy.

The porosity of initial and after gradual degradation PPF samples were examined by low temperature N_2_ adsorption experiments (Figure 3C). The presented results confirm that they showed similar approximate type-I Langmuir isotherms. Only for the samples after the last ST-4 test (immersion in boiling water) did the isotherm become type-II due to an increase in mesoporosity [28,29]. The specific surface area (SBET) of the initial PPF is 255 m^2^ g^−1^. The measured area changes only slightly (i.e., equals 256, 246, and 237 m^2^ g^−1^) when treated with increasingly harsh ST-1, ST-3, and ST-4 conditions, respectively.

The pore size distribution data (inset in Figure 3C) indicate that only the PPF after immersion in boiling water increase slightly the pore diameter to 1.16 nm. The other samples have a similar pore diameter of 1.08 nm, which is in good consistency with the value of 1.02 nm based on crystallographic data. The similar N_2_ adsorption isotherms and pore size indicate that tests under ST-1 to ST-3 conditions have little effect on the pore structure of PPF nanosheets.

To verify if surface functionalities undergo some degradation processes, Fourier transform infrared (FTIR) spectroscopy analyses were performed. As shown in Figure 3D the spectral changes are negligible, and precise analysis is possible only for differential spectra. In general, the PPF spectrum consists of five characteristic bands that appeared from: (i) ν(OH) and ν(NH) stretching vibrations at the range of 3700–2500 cm^−1^; (ii) aromatic rings and ν(C=N) stretching near 1600 cm^−1^; (iii) asymmetric and symmetric stretching vibrations of carboxyl anions, respectively, at ~1550 and ~1380 cm^−1^; and (iv) the ν(C-N) vibration in this case at 1000 cm^−1^. These functionalities remain stable while samples were treated under ST-1 to ST-3 conditions. However, after immersion in boiling water, most probably via H_2_O molecules being incorporated into the 2D structure, thus changing in electron density distribution, some spectral changes can be observed. The blue shifting of all mentioned above bands is typical for the inductive effect of electron-rich molecules interacting, e.g., via hydrogen bonding, and means that bond lengths decrease. This assumption is covered by increasing in intensity of ν(OH) band after trapping water molecules.

The abovementioned results can be summarized that we have obtained stable PPF, with maintained structure and surface chemistry, and slow, easy to be controlled rate of degradation. Another requirement to be met for DDS is their adsorption/desorption ability.

To estimate the adsorption capacity of PPF, adsorption and desorption studies of model compounds—methylene blue and DOX were performed (Figure 4). We proved that about 20% of the initial concentration of both adsorbates was adsorbed by PPF after 96 h incubation at 4 °C. After the precipitate separation, the samples were subjected to desorption studies. Since MB is well solubilized in water, its desorption was faster during the initial 10 h (80% of loaded MB), and then the desorption rate slowed down, reaching approximately 100% after 27 h. In contrast, doxorubicin desorption is much slower from the beginning of the experiment, reaching only around 35% of adsorbed DOX after 27 h.

During other experiments, where desorption has been initiated by drastically changing the pH via addition of a small portion (50 µL) of 2 M HCl (Figure 4C), the huge desorption acceleration relative to desorption in water was observed in both cases. Moreover, stepwise desorption occurred. For MB—highly soluble in water, only 3 portions of HCl were necessary to release the dye totally. Contrary for DOX, even 7 doses of HCl caused only 60% release. Interestingly, through better PPF solubility in an acidic environment we have obtained additional effect—“irregular shape step”. Just after HCl addition, DOX was removed from the surface, and next re-adsorption occurred due to the fact that H_2_TCPP, as a week acid, can be formed.

Then, we aimed to test adsorption and desorption processes with our drug of choice—Sorafenib. While MB and DOX adsorb efficiently from water solutions, SOR adsorption was negligible due to its extremely low solubility in water. We decided to test SOR adsorption on PPF from four different (from 0 to 100%) ethanol/water solutions harnessing innovative sonochemical methods. The best adsorption capacity was observed from 60% water-ethanol solution, and the adsorption rate reached 84% after 96 h (Figure 5A). Moreover, this sample exhibits the highest dispersion of SOR on the PPF surface (Figure 5D). We have also confirmed with SEM images the structure of obtained PPF (Figure 5B), as well as deposition of SOR on the PPF in two differently sized forms: aggregates (Figure 5C) or homogenous layer (Figure 5D). We hypothesized that these forms will also result in different desorption rates as the drug must be slowly desorbed from large aggregates in contrast to fast desorption from well dispersed layers. Thus, we have labelled the obtained systems: SR—Slow Released and FR—Fast Released, respectively. Nevertheless, the regular desorption studies were impossible to perform in any EtOH-free solution because of SOR insolubility.

Next, we aimed at confirming the efficacy of obtained DDS directly with in vitro studies. Rat hepatoma cells (ATCC^®^, CRL 1601) were exposed to growing concentrations of SOR, SR-SOR@PPF, and FR-SOR@PPF for 24 and 72 h. The viability of cells was calculated based on the results of MTT test and related to control cells—Figure 6A.

Based on the above results, it is evident that SOR itself has been cytotoxic for hepatoma cells in a concentration- and time-dependent manner. However, the SR-SOR@PPF and FR-SOR@PPF delivery systems proved to be more effective at almost every concentration tested. These differences can be connected to the SOR adsorption method—if molecules are aggregated on the PPF surface, they are gradually but slowly released. This phenomenon is also reflected with the reduction of effective concentration (EC50)—we have summarized these values in Table 1.

The sensitivity of rat hepatoma cells to sorafenib is similar to those reported in the literature for different liver cancer cell lines. After 24 and 48 h of exposure to sorafenib, the IC50 values in HepG2 cells were 19.5 ± 1.4 and 12.0 ± 3.1 μM, respectively, and in Huh7 cells 15.5 ± 4.4 and 11.3 ± 1.4 μM, respectively [30,31,32]. Sorafenib adsorption on PPF clearly increases its bioavailability. Therefore, the EC50 value decreases almost 10 times to 1.6 μM for fast-release DDS after 24 h and to 1.81 μM for slow-release DDS after 72 h.

The multiple molecular targets of SOR (the serine/threonine kinase Raf and receptor tyrosine kinases) contribute to anti-proliferative activity in liver cancer cell lines and increased cell death directly through downregulation of the Ras/Raf/Mek/Erk signaling pathway (Figure 6B). To confirm that the downstream pathway kinases inhibition underlies the decrease in hepatoma cell viability, we determined the ERK 1/2 phosphorylation level. Our results demonstrated that both SR-SOR@PPF and FR-SOR@PPF acted through multiple kinases inhibition—Figure 6C. Noteworthy, both DDS are more effective in kinase inhibition than SOR itself—FR-SOR@PPF at the lowest concentrations (0.75 μM and 1.5 μM) is more effective after 24 h by approximately 40%. SR-SOR@PPF exhibits a similar increase in efficacy after 72 h.

We can conclude that tested DDS are promising nanodevices with controlled release of SOR to be successfully applied in hepatoma treatment. Therefore, the animal study was performed to confirm that conclusion further.

In the whole in vivo study group (45 rats), the implantation of the hepatoma cells was successful and the presence of tumor was histologically confirmed (Figure 7). Then, SOR alone or two types of SOR nanocarriers (SR-SOR@PPF and FR-SOR@PPF) were implemented. After the second surgery, one rat from the placebo group died due to massive bleeding as a consequence of the sparing resection (which was diagnosed post-mortem). Besides, there were no other incidents which affected the health of the animals. No abnormal changes in behavior were observed, and no additional treatment was administered. None of the other animals have been disqualified from the study. The mean tumor size after the treatment was 3.64 (± 0.24) mm, 2.62 (± 0.27) mm and 1.73 (± 0.36) mm in the C, SR and FR groups, respectively. Animals treated with SR-SOR@PPF and FR-SOR@PPF demonstrated significantly lower tumor size at the end of the study. The outcome was slightly improved, compared to the control, among the FR-SOR@PPF group (*p* = 0.004) than the SR-SOR@PPF (*p* = 0.043) (Figure 8A,B).

The results of the final surgical examination are shown in Table 2 (with intraoperative documentation shown in Figure 8). The reduction of metastatic potential was observed in both groups in which Sorafenib carriers were administered. Regional metastasis appeared less frequently in the SR (odds ratio (OR): 0.15, *p*-value = 0.040) and FR rats (OR: 0.36, *p*-value = 0.048) when compared to the control. Similar results were observed for distant metastases and recurrences in the postoperative scar comparing test groups with control. Among the SR rats, OR of distant metastasis appearance and scar recurrence was 0.26 (*p* = 0.009) and 0.38 (*p* = 0.043), respectively. In the FR group, these parameters were slightly worse with 0.56 OR (*p* = 0.165) for presence of distant involvement, and 0.63 OR (*p* = 0.096) for recurrence within the scar.

In addition, at the end of the study, the general evaluation of the surgery site was performed. Structural changes (e.g., surface folding, nodules, telangiectasias), abdominal adhesions and degree of the carrier degradation were noted during the autopsy. The abnormalities in the liver structure and appearance were observed similarly common in all groups, despite potentially more harmful treatment in the animals in which SOR was released. OR for the telangiectasias or structural changes was 0.53 and 0.67 in the SR and FR rats, respectively. No significant differences were found in comparison with the control group (*p*-value > 0.05). Only two cases of abdominal adhesions in the control group and three cases in both carrier-treated groups were observed at the end of the study. No obstruction, peritonitis, or intestinal necrosis were observed in any of the animals, and taking this factors into account, the groups also did not differ significantly (*p*-value > 0.05).

## 3. Materials and Methods

Synthesis of PPF: The synthesis of porphyrin based 2D MOFs was performed using procedure similar to method presented by Zhao et al. [20]. Briefly, Zn(NO_3_)_2_ 6H_2_O (POCh, 0.250 g) and TCPP (tetrakis(4-carboxyphenyl)porphyrin), (TCI chemicals, Portland, OR, USA, 0.079 g) in a solvent mixture of DMF (N,N-Dimethylformamide, ACROS Organics™,Geel, Belgium)/EtOH (POCh) 99.8% (15 mL/7.5 mL) were placed in a 30 mL glass vial and mixed for about 15 min until total dissolution of both reagents. Then, the vial was placed into oven at 80 °C for 24 h. Purple precipitate was then obtained by filtration on nylon membrane filters (pore size 0.45 μm). The product was washed with 5 mL mixture of DEF (N,N-Diethylformamide) and EtOH (2:1), and 3 times with 10 mL EtOH. The solid was then evacuated at 50 °C under vacuum for 24 h.

Sonochemical drug deposition: Sorafenib tosylate EtOH/H_2_O solution was prepared in two variants differing in EtOH concentration—60 and 40%, respectively, for fast-release (FR) and slow-release (SR). To 20 mL of each solution, 20 mg of PPF was added, and the mixture was placed in the fridge. Mixture was continuously sonicated in the cycle: ultrasounds (2 s) and pause (18 s). The total adsorption time was 96 h.

Methylene blue (MB) or doxorubicin (DOX) solution (1 mg/mL) was prepared. To 20 mL of this solution, 20 mg PPF was added, and the mixture was placed in the fridge. Mixture was continuously sonicated in the cycle: ultrasounds (2 s) and pause (18 s). The total adsorption time was 96 h.

Additionally, for MB and DOX, the desorption studies were performed. Twenty mg of MB and DOX@PPF was placed in 20 mL of water, and kinetic of drug release was measured (at 25 °C). In order to prove the acidic environment influence on drugs release kinetic, the separate experiments were carried out. Every 30 min, a portion of 50 µL HCl (2 M) was added.

PPF degradation in buffers: approximately 7 mg of PPF was diluted in 100 mL of solution (water; phosphate buffer 0.05 M pH = 6.24, 7.0, 7.4; phosphate buffer 0.005 M, pH = 7.0 and 0.0005 M, pH = 7.0; acetate buffer pH = 5.5). 50 or 100 μL aliquots were taken from the obtained solutions and transferred to spectrometric cuvette in different time points, then the solution was immediately diluted to 3 mL, and the UV-Vis spectrum was measured.

PPF degradation under various conditions: we examined the following conditions for degradation/stability tests: (i) ST-1 at 80 °C and 90% RH, (ii) ST-2 at 4 °C and immersion in H_2_O, (iii) ST-3 at 20 °C and immersion in H_2_O, and (iv) ST-4 at 100 °C and immersion in H_2_O.

ST-1 conditions were achieved in desiccator with saturated solution of KCl. After 24 h of temperature and humidity stabilization, glass vial with 0.04 g of sample was placed in desiccator for the next 24 h. After that, time samples were evacuated at 50 °C and vacuum for 24 h to yield activated sample.

For ST-2 and ST-3—0.04 g of sample was transferred to glass vial and then 40 mL of water was added. Glass vial was mixed continuously with laboratory shaker. After 24 h, precipitate was obtained by filtration through nylon membrane filters (pore size 0.8 μm). The solid was then evacuated at 50 °C and vacuum for 24 h to yield activated sample.

In the case of ST-4—0.04 g of sample was transferred to a glass round flask and then 40 mL of water was added. The flask was installed in the heating mantle. Temperature of heating solution was kept around 100 °C for 24 h. After 24 h, precipitate was obtained by filtration through nylon membrane filters (pore size 0.8 μm). The solid was then evacuated at 50 °C and vacuum for 24 h to yield activated sample.

### 3.1. In Vitro Studies

Cell culture: human dermal fibroblasts (HDF) were cultured in F-12 medium supplemented with 10% FBS and 100 IU/mL of penicillin and 50 µg/mL streptomycin (Sigma-Aldrich, Darmstadt, Germany). Rat hepatoma McA-RH7777 cell line (ATCC^®^, CRL 1601) was cultured in vitro according to the manufacturer protocol. Shortly, the culture medium consisted of DMEM-HG and 10% FBS with 100 IU/mL of penicillin and 50 µg/mL streptomycin (Sigma-Aldrich, Germany). The cells were grown at 37 °C under a humidified atmosphere containing 5% CO_2_. The cells were passaged using 0.25% trypsin-EDTA solution (Sigma-Aldrich) when reaching 70–80% of confluency.

Viability assays: cell viability was determined after 24 and 72 h using the MTT (3-(4,5-dimethylthiazole-2-yl)-2,5-diphenyl tetrazolium bromide (Sigma-Aldrich, Germany) assay. Fibroblasts and hepatoma cells were seeded in a 48-well culture plates at a density 1 × 10^4^ cells/well. The cells were cultured for 24 and 72 h with different concentrations of sorafenib tosylate (SOR), SOR@PPF, PPF, and without any compounds (control). After the respective time, the culture medium was discarded and 300 µl of MTT (1 mg/mL, Sigma-Aldrich) solution in a suitable culture medium without phenol red was added to each well. After 1 h of incubation at 37 °C in water bath, the solution was aspirated, 500 μL of dimethyl sulfoxide (DMSO, 100%, Sigma-Aldrich) was added to each well and the plates were shaken for 10 min. The absorbance was measured at the wavelength of 570 nm using a microplate reader (BioTek, Winooski, VT, USA).

ERK 1/2 phosphorylation assay: the level of phosphorylated ERK 1/2 (extracellular signal-related kinase) was used as a common end-point measurement for the activation of G protein coupled receptors, e.g., PDGFR and VEGFR. ElisaOne™ Erk1/2 Assay Kit (TGR Biosciences, Thebarton, Australia) was used for analysis according to the manufacturers’ protocol. At the completion of cell treatment, the hepatoma cells were lysed and 50 μL samples of lysates were mixed with antibody mix and incubated for 1 h at room temperature with shaking (~200 rpm). The immuno-complexes were detected on the basis of enzymatic reaction and measured spectrophotometrically at 405 nm. Protein concentration in cell lysates was assayed with the Bradford method. The level of ERK 1/2 phosphorylation was normalized to protein content in samples.

### 3.2. In Vivo Studies

The in vivo study was performed and approved by the local ethical committee associated with the University of Science and Technology in Bydgoszcz (Poland). Furthermore, the procedures were conducted, and the number of animals used in the experiment was reduced, in accordance with the guidelines of the European Union (Directive 2010/63/EU; Consolidated version: 26 June 2019) and other leading standards [33].

In total, 50 Sprague Dawley rats, 4–8 weeks old (AnimaLab, Poznan, Poland) were subjected to study. A whole procedure of tumor implantation was prepared in accordance with the procedure presented by the other research groups [34,35]. In five rats, McA-RH7777 cells (ATCC^®^, CRL 1601) were injected subcutaneously (5 mln each). These rats were donors for the rest of the animals. The 45 rats were randomly divided into 3 groups: C—control group, in which Sorafenib solution (7.5 mg/mL) was administered (*n* = 15); SR-SOR@PPF—slow-releasing Sorafenib carriers (*n* = 15); FR-SOR@PPF—fast-releasing Sorafenib carriers (*n* = 15). Two weeks after the cell injections in the donor group, the tumors were excised, divided into 1 mm^3^ cubes and instantly implanted into the recipients’ median lobe of the liver. After another two weeks, a sparing resection without margin (a part of the tumor about 5 mm in size was not excised in each case) of the cancerously involved site was performed. The operation was preceded by an ultrasonographic examination in which tumor growth was confirmed. After the surgery the tissues were histologically assessed. In the control, SR and FR groups Sorafenib (solution or carriers) were placed within the postoperative space. Sorafenib dose was calculated in accordance with the current clinical standard in the hepatoma treatment and translated according to the animal’s body weight [36]. Four weeks after the surgery, the rats were euthanized and the post-mortem surgical examination was performed.

Statistical analyses: all experiments were performed at least in triplicate. Statistical analyses were applied when possible. The results are presented as mean ± SD. Results from animal studies were analyzed using Statistica 13.3 software (TIBCO Software Inc, Carlsbad, CA, USA). The *t* test was applied to compare differences in tumor size between groups at the end of the study. Postoperative results, including metastatic appearance, hepatic damage and abdominal adhesions were compared using Fisher exact probability test. A *p*-value below 0.05 was considered significant.

## 4. Conclusions

Concluding this experimental part, the capability of novel SOR-releasing nanocarriers to reduce liver cancer growth, metastasis, and reoccurrence was confirmed. Drug-releasing SOR nanocarriers induce alterations in the liver structure and adhesions to an extent similar to SOR alone. Notably, using two distinct DDS with the controllable release, we can obtain different clinical outcomes. FR-SOR@PPF acts efficiently, fast, limits the tumor growth, and decreases lymph nodes involvement and distant metastases by 23% and 8% relative to control. On the other hand, SR-SOR@PPF acts slower. After treatment, the mean tumor size is smaller than in the control group but larger than in animals administered FR-SOR@PPF. It most efficiently decreases the lymph nodes involvement and distant metastases. These differences in tumor treatment results can be ascribed inter alia to the presence of cancer stem cells (CSCs). Liver CSCs represent an atypical population of tumor cells crucially involved in drug resistance. It is commonly accepted that existing therapeutic strategies mostly focus on the inhibition of tumor growth, resulting in the death of bulk tumor cells. Hence, a small group of resistant CSCs remains in the niche and contributes to local tumor recurrence as well as to distant metastases [37].

Altogether, we proved that SOR adsorption into the PPF structure could solve the key disadvantages that limit drug efficacy, bioavailability, and patient outcomes. Based on our results, it is justified to expect that an appropriate mixture of both FR- and SR-SOR@PPF would be beneficial new DDS for the most efficient hepatoma treatment, including activity towards cancer stem cells. This issue deserves more attention and further detailed studies.

## Figures and Tables

**Figure 1 ijms-22-11161-f001:**
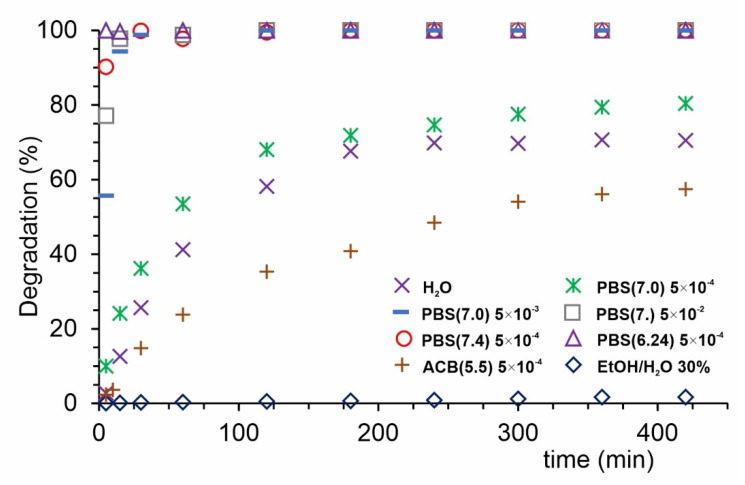
PPF degradation kinetics in various solutions.

**Figure 2 ijms-22-11161-f002:**
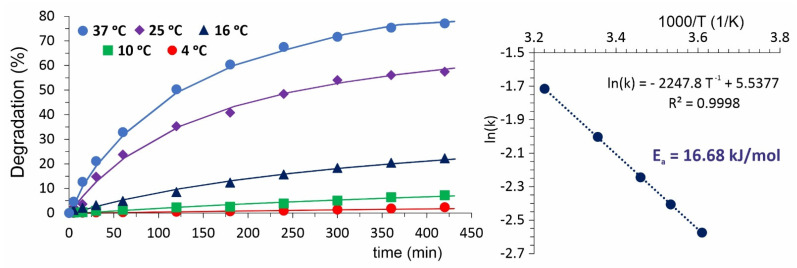
Influence of temperature on PPF degradation kinetics.

**Figure 3 ijms-22-11161-f003:**
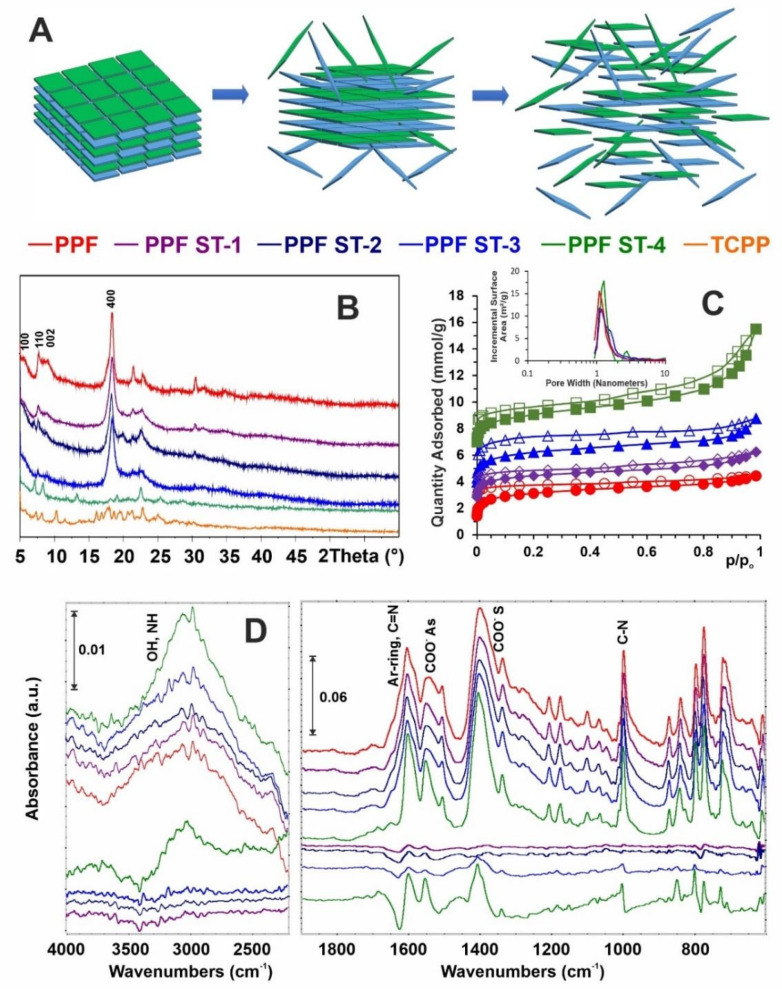
Characteristics of PPF solids remaining after degradation. (**A**) Degradation scheme; (**B**) XRD patterns; (**C**) low temperature N_2_ adsorption isotherms (note that isotherms for PPF ST-1, PPF ST-3, PPF ST-4 have been shifted for clarity by 1, 3, and 6 mmol/g, respectively); (**D**) FTIR analysis results.

**Figure 4 ijms-22-11161-f004:**
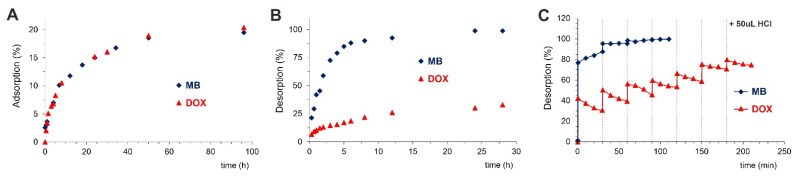
(**A**) Adsorption (at 4 °C) and (**B**) desorption (at 25 °C) of model compounds—methylene blue (MB, blue curves) and doxorubicin (DOX, red curves) from PPF structure, and (**C**) effect of strong acid on the desorption of MB and DOX; note that each vertical dotted line denotes the moment of addition of 50 µL 2 M HCl to the solution.

**Figure 5 ijms-22-11161-f005:**
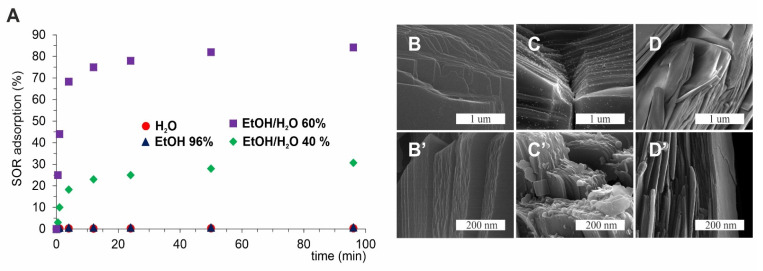
SOR adsorption on PPF. (**A**) Time dependent adsorption from different solutions; (**B**–**D**,**B’**–**D’**) SEM pictures of PPF, SR-SOR@PPF, and FR-SOR@PPF, respectively.

**Figure 6 ijms-22-11161-f006:**
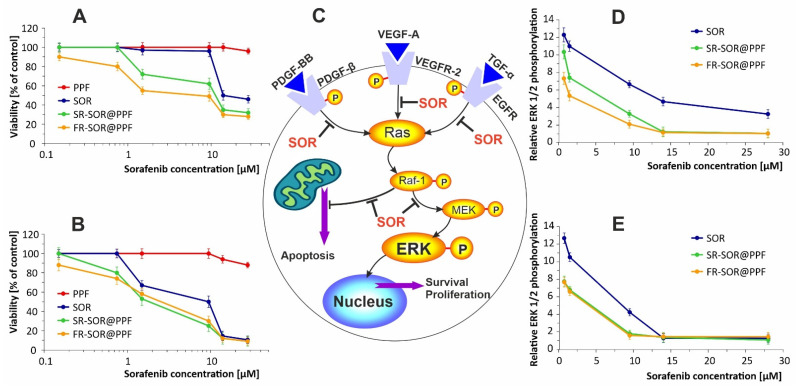
Viability of hepatoma cells after 24 h (**A**) and 72 h (**B**) of treatment with SOR, SR-SOR@PPF, and FR-SOR@PPF delivery systems. (**C**)—schematic representation of SOR inhibited pathways. Relative ERK 1/2 phosphorylation level in hepatoma cells after 24 h (**D**) and 72 h (**E**) treatment. The values are normalized to protein content; note that the phosphorylation level of untreated cells was 12.3 ± 0.40 after 24 h, and 12.6 ± 0.38 after 72 h.

**Figure 7 ijms-22-11161-f007:**
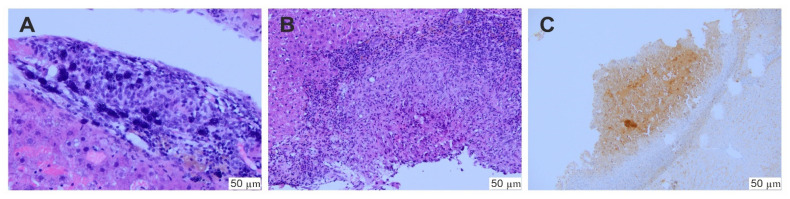
Histological assessment of the biopsy. (**A**) Hepatoma cells, (**B**) a minor tumor with severe fibrosis and inflammation; (**C**) hepatoma confirmation—immunochemical reaction for alpha-fetoprotein.

**Figure 8 ijms-22-11161-f008:**
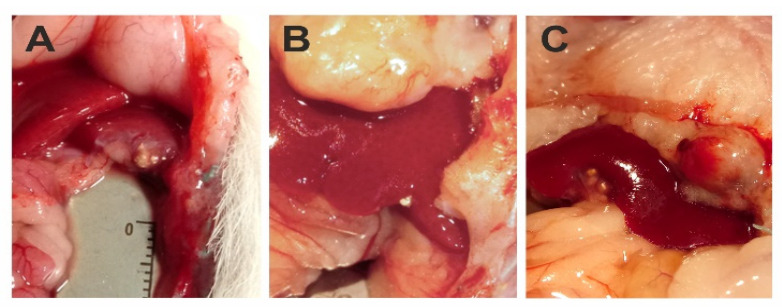
The tumor identification: (**A**)—before the drug administration, (**B**)—after FR-SOR@PPF treatment, (**C**)—regional (pre-hepatic space) metastatic tumor in control animal.

**Table 1 ijms-22-11161-t001:** Effective concentration (EC50) for anti-proliferative activity of sorafenib and SR-SOR@PPF or FR-SOR@PPF delivery systems.

Time	EC50 [μM]		
	SOR	SR-SOR@PPF	FR-SOR@PPF
24 h	13.8 ± 0.8	11.6 ± 1.0 ****	1.6 ± 0.04 ****
72 h	10.0 ± 0.7	1.8 ± 0.1 ***	2.9 ± 0.08 ***

Calculated *p* value between SOR and SR-SOR@PPF or FR-SOR@PPF was marked with asterisks (* for *p* ≤ 0.05, ** for *p* ≤ 0.01).

**Table 2 ijms-22-11161-t002:** Summarized results of the surgical examination of rats after SOR and SOR-releasing nanocarriers administration.

Group (*n*)	Tumor Size, Median (mm)	Lymph Nodes Involvement (%)	Distant Metastasis (%)	Recurrence in Postoperative Scar (%)	Adhesions (%)	Telangiectasias on the Surface of the Liver (%)	Change in the Structure of the Liver (%)
Control (14)	3.64 ± 0.24	50.0	21.4	28.6	35.7	28.6	14.3
SR (15)	2.62 ± 0.27	13.3	6.7	13.3	33.3	20.0	20.0
FR (15)	1.73 ± 0.36	26.7	13.3	20.0	40.0	40.0	20.0

## Data Availability

The data will be available from authors upon reasonable request.

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
