# Peer review of "Porphyrin Based 2D-MOF Structures as Dual-Kinetic Sorafenib Nanocarriers for Hepatoma Treatment"

_ijms, 2021, doi:10.3390/ijms222011161_

Round 1

Reviewer 1 Report

The authors reported the porphyrin-based, paddle-wheel framework (PPF) as dual-kinetic Sorafenib (SOR) nanocarriers against hepatoma. B, the augment of SOR anticancer effects was realized by controlling the PPF degradation and size of adsorbed SOR deposits. Furthermore, the in vivo studies proved that fast-released SOR@PPF reduces the tumor size considerably, and the slow-released SOR@PPF prevents much better from lymph nodes involvement and distant metastases. This work is very meaningful for the development of MOF based drug carriers. I recommend its publication after addressing the following comments.

  1. Please supply the formula of PPF and the ecotoxicity test of metal ions and porphyrin ligands.
  2. Please supply the simulated powder XRD results in Figure 3b.
  3. As shown in Figure 3c, the N2 adsorption isotherms of PPFs under different conditions were quite different (from ca. 4 mmol/g to ca.14 mmol/g), while the authors claimed that ‘The measured area changes only slightly, 255, 256, 246, and 237 m2/g’. the authors are suggested to check the adsorption data carefully.
  4. Some new references about MOFs should be cited, such as Chem. Soc. Rev., 2021, 50, 5706-5745. Chem. Soc. Rev., 2018, 47, 2130-2144. ACS Materials Letters, 2021, 3, 64-68.

Reviewer 2 Report

In this work, Porphyrin Based 2D-MOF Structures as Dual-Kinetic Sorafenib 2 Nanocarriers for Hepatoma Treatment, Bieniek and coworkers studied the uptake and release of Sorafenib in vitro and in vivo in human dermal fibroblasts (HDF) and Rat hepatoma McA-RH7777 cell lines using a Porphyrin-based, paddle-wheel framework (PPF). As a general suggestion, while several MOFs described in 77-104 (and many more) are studied as drug delivery systems, especially for drugs that show poor solubility in water, and their stability in the aqueous medium is a challenge. The sloughing of the PPF as a mechanism of drug release is an interesting one but has limitations. As a system free of metals, covalent organic frameworks provide a water-stable platform for drug delivery. Furthermore, the H-binding sites between the drug (such as Quercetin) and the COF are better understood there.

A couple of typos in lines 142 and 350 can be fixed during the editing process.

Overall, the experiments are thorough, and the manuscript has been drafted well. I recommend it for publication in the International Journal of Molecular Sciences.
